# Pixel Size and Revisit Rate Requirements for Monitoring Power Plant CO$_2$ Emissions from Space

**Tim Hill** [1,*] **and Ray Nassar** [2]

1   Department of Applied Math, University of Waterloo, Waterloo, ON N2L 3G1, Canada
2   Climate Research Division, Environment and Climate Change Canada, Toronto, ON M3H 5T4, Canada
*   Correspondence: tghill@uwaterloo.ca

**Abstract:** The observational requirements for space-based quantification of anthropogenic CO$_2$ emissions are of interest to space agencies and related organizations that may contribute to a possible satellite constellation to support emission monitoring in the future. We assess two key observing characteristics for space-based monitoring of CO$_2$ emissions: pixel size and revisit rate, and we introduce a new method utilizing multiple images simultaneously to significantly improve emission estimates. The impact of pixel size ranging from 2–10 km for space-based imaging spectrometers is investigated using plume model simulations, accounting for biases in the observations. Performance of rectangular pixels is compared to square pixels of equal area. The findings confirm the advantage of the smallest pixels in this range and the advantage of square pixels over rectangular pixels. A method of averaging multiple images is introduced and demonstrated to be able to estimate emissions from small sources when the individual images are unable to distinguish the plume. Due to variability in power plant emissions, results from a single overpass cannot be directly extrapolated to annual emissions, the most desired timescale for regulatory purposes. We investigate the number of overpasses required to quantify annual emissions with a given accuracy, based on the mean variability from the 50 highest emitting US power plants. Although the results of this work alone are not sufficient to define the full architecture of a future CO$_2$ monitoring constellation, when considered along with other studies, they may assist in informing the design of a space-based system to support anthropogenic CO$_2$ emission monitoring.

**Keywords:** CO$_2$ emissions; satellite; coal; climate change; anthropogenic; power plant; greenhouse gases

## 1. Introduction

CO$_2$ emissions from fossil fuel combustion have elevated atmospheric CO$_2$ above pre-industrial levels, altering the radiative forcing of the atmosphere and the global climate. The United Nations Framework Convention on Climate Change (UNFCCC) Paris Agreement aims to address this issue by coordinating contributions to reduce greenhouse gas (GHG) emissions at various scales by nations and acknowledging the role of subnational and non-state actors. However, to better manage emissions, improved and/or complementary methods of quantifying them are needed at all spatial scales from the national level down to the facility level. National GHG emissions are reported under the UNFCCC according to protocols of the Intergovernmental Panel on Climate Change Task Force on National Emission Inventories [1]. Approved methods are mostly activity-based, assuming some quantity of emissions for a given quantity of fuel burned or other proxy, and seldom deal with atmospheric GHG measurements. While constructing a system of in situ measurements to monitor anthropogenic CO$_2$ emissions may be possible at some scales for select locations, it would be a major logistical challenge to implement globally, while satellites are frequently used as a means of acquiring global observations.

Although no satellites have yet been launched specifically for anthropogenic $CO_2$ emissions monitoring, they are now being considered for this purpose [2] and could play a key role in an Integrated Global Greenhouse Gas Information System (IG3IS) [3]. Nassar et al. [4] show that monitoring $CO_2$ emissions at the scale of individual mid to large sized (>10 Mt yr$^{-1}$) fossil fuel burning power plants should be possible with multiple satellites. The preferred architecture for such a constellation is now being considered by space agencies, Committee on Earth Observations Satellites (CEOS), the Coordination Group for Meteorological Satellites (CGMS) and other international groups. Bovensmann et al. [5] first demonstrated with simulated data, the importance of spatially imaging column-averaged $CO_2$ mole fraction (XCO$_2$) plume enhancements for quantifying point source emissions with a pixel size of $2 \times 2$ km$^2$ as proposed for CarbonSat. Nassar et al. [4] used XCO$_2$ observations from NASA's Orbiting Carbon Observatory 2 (OCO-2) [6], which has a narrow swath consisting of eight parallelogram-shaped $\leq 1.29 \times 2.25$ km$^2$ footprints, to quantify daily emissions from five power plants. For three U.S. power plants, the estimated emissions were verified to be within 1%, 4% and 17% of daily emissions reported by the U.S. Environmental Protection Agency (EPA), which enabled application of the method to power plants in India and South Africa, which have less detailed or less reliable available information on $CO_2$ emissions.

The present study does not prescribe a detailed design for a future $CO_2$ monitoring constellation, which would need to function for emission estimates at various scales, but simply investigates pixel size, a multiple image averaging method, and revisit rate in the context of quantifying power plant $CO_2$ emissions. The impact of varying pixel size is explored using simulated data that contain noise and various types of biases, and the accuracy and precision of square pixels is compared to that of rectangular pixels with equal area and a 3 to 1 aspect ratio. A method is introduced and evaluated that averages multiple images with varying wind directions to obtain averaged enhancements nearly free of background noise and with a higher effective resolution. The number of required revisits to constrain annual power plant $CO_2$ emissions with a given percent uncertainty is explored with a statistical approach based on the temporal variability in U.S. coal power plant $CO_2$ emissions using the most recent three years of reported emission data from the 50 highest emitting US power plants, and a compact relationship for the number of overpasses given a desired precision is derived.

## 2. Methods

### 2.1. Simulation Domain

We assessed the ability to quantify $CO_2$ emissions from power plants by simulating XCO$_2$ enhancements over a square 60 km $\times$ 60 km region, which we refer to as the field of view (FoV). The domain was subdivided into a regular 50 m $\times$ 50 m grid. Data were averaged to simulate observations with square pixels with dimensions $2 \times 2$ km$^2$, $4 \times 4$ km$^2$, $7 \times 7$ km$^2$, and $10 \times 10$ km$^2$ (see Figure 1). Additionally, rectangular pixels with dimensions $2.3 \times 7$ km$^2$ were investigated. These rectangular pixels had an area almost identical to that of the $4 \times 4$ km$^2$ pixels, but we expect a disadvantage due to their 3:1 aspect ratio (see Figure 2).

### 2.2. Gaussian Plume Model

XCO$_2$ enhancements are simulated using a vertically-integrated Guassian plume model as in Nassar et al. [4] that is based on a slightly modified set of equations from those used by Bovensmann et al. [5]. The total column averaged enhancement above the background concentration is computed as

$$V(x,y) = \frac{F}{\sqrt{2\pi}\sigma_y(x)u} e^{-\frac{1}{2}\left(\frac{y}{\sigma_y(x)}\right)^2},$$

(1)

where

$$\sigma_y(x) = a\left(\frac{x}{x_0}\right)^{\gamma}.$$

(2)

$V(x, y)$ is the $CO_2$ vertical column density in grams per square meter ($\mathrm{g\,m^{-2}}$) at coordinates $(x, y)$, where $x$ is the distance in meters downwind of the point source, and $y$ is the distance in meters perpendicular to the wind. $F$ is the source emission rate in grams per second ($\mathrm{g\,s^{-1}}$), $u$ is the wind speed in meters per second ($\mathrm{m\,s^{-1}}$), $a$ is the atmospheric stability parameter as determined from the Pasquill–Gifford stability class [7], $x_0 = 1000$ m is the characteristic length of the across wind diffusion, and $\gamma = 0.894$ is a constant. $\sigma_y(x)$ determines the rate of diffusion perpendicular to the wind. The atmospheric stability parameter $a$ may be more accurately described as the half-width of the plume at $x = x_0$, however, we follow the convention of the literature. These equations assume the motion in $x$ is dominated by advection due to the wind, the flow is not turbulent at the scales it is modelled at, and there are no boundary layer effects.

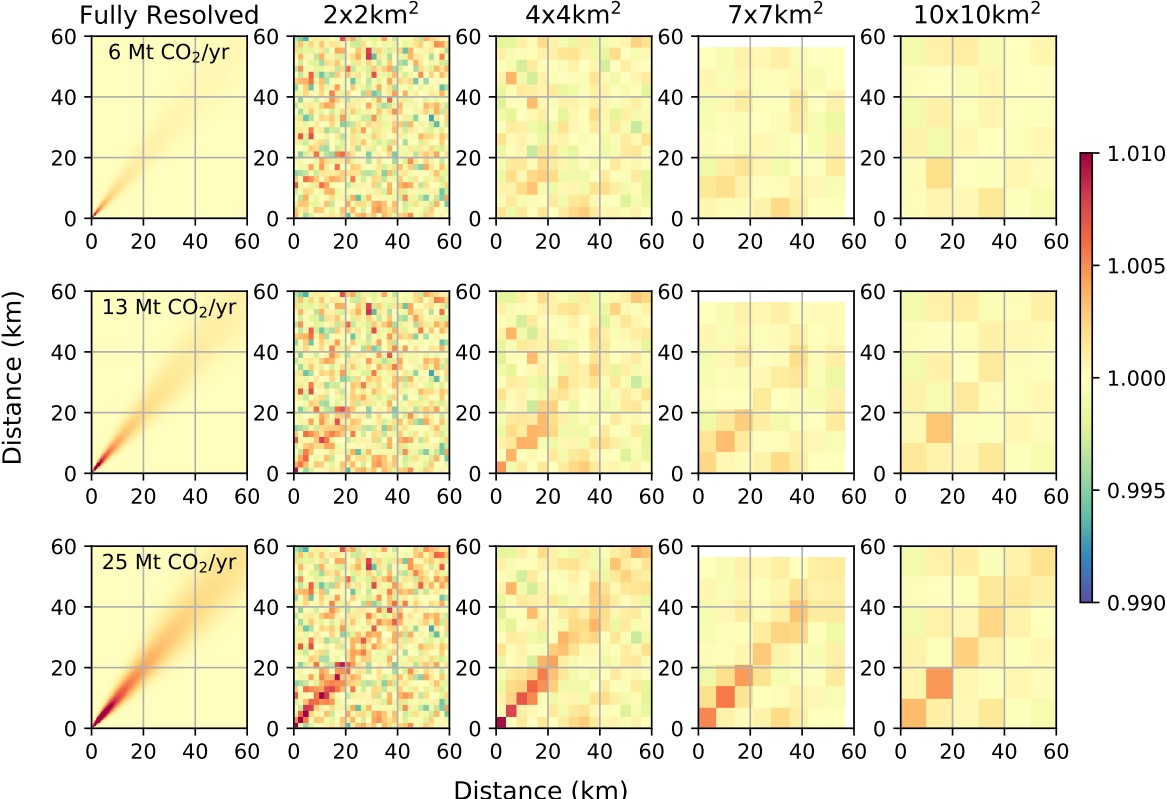

**Figure 1.** Model and simulated column-averaged $CO_2$ mole fraction ($XCO_2$) observations relative to a constant 400 ppm background for 3 source emission rates and 4 pixel sizes. The first column shows the simulated Gaussian plume at 50 m resolution for each power plant emission rate against a clean background. Columns 2, 3, 4, and 5 show the simulated observations averaged to pixels of size $2 \times 2$ km$^2$, $4 \times 4$ km$^2$, $7 \times 7$ km$^2$, and $10 \times 10$ km$^2$, respectively. The top row shows the simulated observations for a source with emission rate 6 Mt yr$^{-1}$, the middle row shows observations for a 13 Mt yr$^{-1}$ source, and the bottom row shows observations for a 25 Mt yr$^{-1}$ source.

Enhancements are computed on the 50 m resolution grid for sources with annual emissions of 6, 13, and 25 Mt $CO_2$ yr$^{-1}$. The largest fossil fuel burning power plants in the world can emit $\sim$30 Mt $CO_2$ yr$^{-1}$. Those in the range of 10–20 Mt $CO_2$ yr$^{-1}$ can be considered mid-sized with the lowest emitting power plant for which emissions were quantified in Nassar et al. [4] emitting 10 Mt $CO_2$ yr$^{-1}$. Past studies suggest that the 6 Mt yr$^{-1}$ source should be difficult to estimate with a difficult to discern plume, the 13 Mt yr$^{-1}$ source should usually be able to be estimated with a weak but visible plume, and the 25 Mt yr$^{-1}$ source should be easy to estimate with a strongly discernible plume [4,5].

For all simulations unless stated otherwise, the model parameters were: $u = 4 \text{ m s}^{-1}$, $a = 156.0 \text{ m}$, $\theta = 45°$ for square pixels and $\theta = 0°$ for rectangular pixels, background $XCO_2 = 400 \text{ ppm}$, and source emission rates are $F = 6, 13, 25 \text{ Mt yr}^{-1}$. Noise strength for the 2 km square pixels is 1 ppm, or 0.25%.

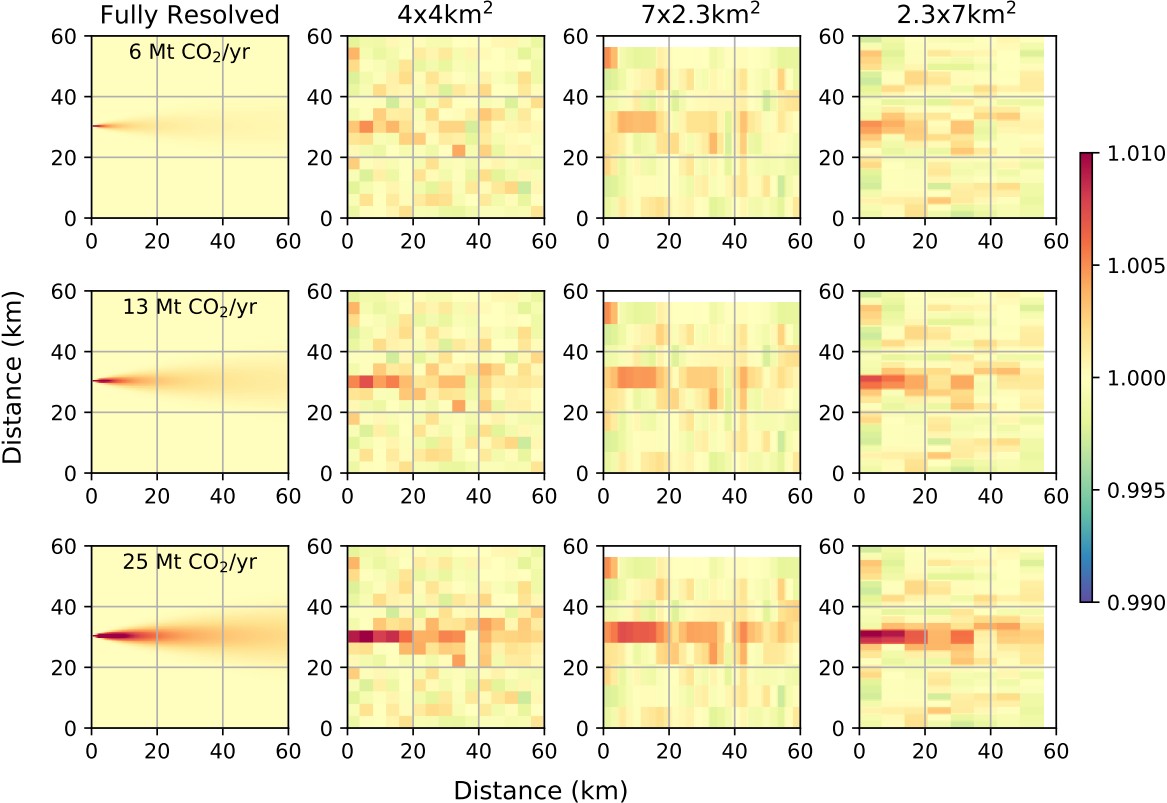

**Figure 2.** Simulated and noisy $XCO_2$ observations relative to a constant 400 ppm background as in Figure 1. Column 2 shows the noisy observations for square $4 \times 4 \text{ km}^2$ pixels. Columns 3 and 4 show the noisy observations for rectangular $7 \times 2.3 \text{ km}^2$ pixels with the long side oriented across the plume (column 3) and along the plume (column 4).

### 2.3. Simulating Instrument Noise

Noise was superimposed over the simulated enhancements to represent random errors that would occur in true observations. The strength of the plume enhancements in comparison to the strength of the noise was the primary limitation in determining how large a power plant must be to accurately estimate its emissions.

The noise or random error was generated following a normal distribution, and the strength of the noise was defined as its standard deviation. The default noise strength was 1 ppm for $2 \times 2 \text{ km}^2$ pixels, representing an instrument with a precision of 1 ppm, or 0.25% at 400 ppm $CO_2$ background. This was similar to the precision requirement of OCO-2 [6]. For a space-based instrument of a fixed size, higher precision is more easily achieved with larger pixels. With larger pixels the instrument collects more light such that the signal $S$ increases proportional to the area $A$ ($S \propto A$), however, noise $N$ also increases in proportion to the square root of the signal, hence $N \propto \sqrt{A}$. The resulting improvement to the signal-to-noise ratio (SNR) is proportional to $\sqrt{A}$. Equivalently, for square pixels the SNR increases linearly with the pixel side length, hence we scaled the precision to be inversely proportional to the pixel side length.

### 2.4. Simulating Biases

Several biases are simulated representing both biases in the observations and errors in the methodology. Instrument biases include: a footprint position dependent bias, retrieved surface albedo errors, and a bias proportional to the gradient of the surface elevation. Methodological biases include: the loss of data in the FoV due to clouds, open water, or an instrument with a narrow FoV; interfering sources not accounted for in the model; incorrect atmospheric stability parameter; and incorrect wind directions. Methods for simulating biases and results for all individual biases are described in the supplementary information.

### 2.5. Emission Estimates

The power plant emissions are estimated from the simulated observations in the same way as in Nassar et al. [4]. The simulated observations containing noise and possible observational biases or methodological errors are compared to theoretical enhancements from the plume model (Equations (1) and (2)). To estimate the emissions, we define the enhancement sensitivity $\alpha$ as

$$\alpha(x,y) = \frac{1}{\sqrt{2\pi}\sigma_y(x)u}e^{-\frac{1}{2}\left(\frac{y}{\sigma_y(x)}\right)^2}. \tag{3}$$

The enhancement sensitivity is computed across the FoV, and a linear least squares fit is carried out between the observations and the sensitivity function to find the estimate $F^*$ that minimizes the squared error between the observations and $\alpha F^*$. The estimate is converted from g s$^{-1}$ to Mt CO$_2$ yr$^{-1}$ for comparison to the simulated emission rate.

### 2.6. Averaging Multiple Images

With a noise strength of 1 ppm at $2 \times 2$ km$^2$ resolution, the enhancements from power plants with emissions less than about 5 Mt CO$_2$ yr$^{-1}$ are indistinguishable from the background noise even for the smallest pixel sizes. For larger pixels, the plume was even harder to distinguish. To account for weak sources of SO$_2$, Fioletov et al. [8] averaged many individual overpasses of SO$_2$ from the Ozone Monitoring Instrument (OMI) to achieve a higher effective resolution and precision than any individual OMI overpass. In this way they were able to isolate power plant plumes that are not detectable in any individual overpasses. We investigate a similar technique for simulated XCO$_2$ observations in this study.

For the purpose of this method, medium-sized $4 \times 4$ km$^2$ square pixels were used with 0.5 ppm noise. The wind speed was $u = 4$ m s$^{-1}$, and the corresponding stability parameter is $a = 156.0$ m. Enhancements were simulated on a 60 km $\times$ 60 km domain at 50 m resolution with a medium size 13 Mt yr$^{-1}$ source in the center of the domain for six different wind directions (Figure 3). The observations for each of the six images were first averaged to the $4 \times 4$ km$^2$ pixel size, and the 0.5 ppm noise was added to simulate observation by such an instrument. The observations were then rotated back so that the plumes are all aligned. At this stage, the 4 km $\times$ 4 km grids for each wind direction are misaligned. To average the enhancements, the observations for each direction must be interpolated onto a uniform grid. We interpolate the enhancements onto the original 60 km $\times$ 60 km grid at 50 m resolution using a nearest neighbour method. Pixels in the corners of the regular grid that were not covered by the rotated grids use the nearest value of the rotated grid. Figure 4 shows the enhancements before and after being interpolated onto the regular grid. Finally, the observations across the images were averaged, resulting in a clear plume outline with reduced background noise.

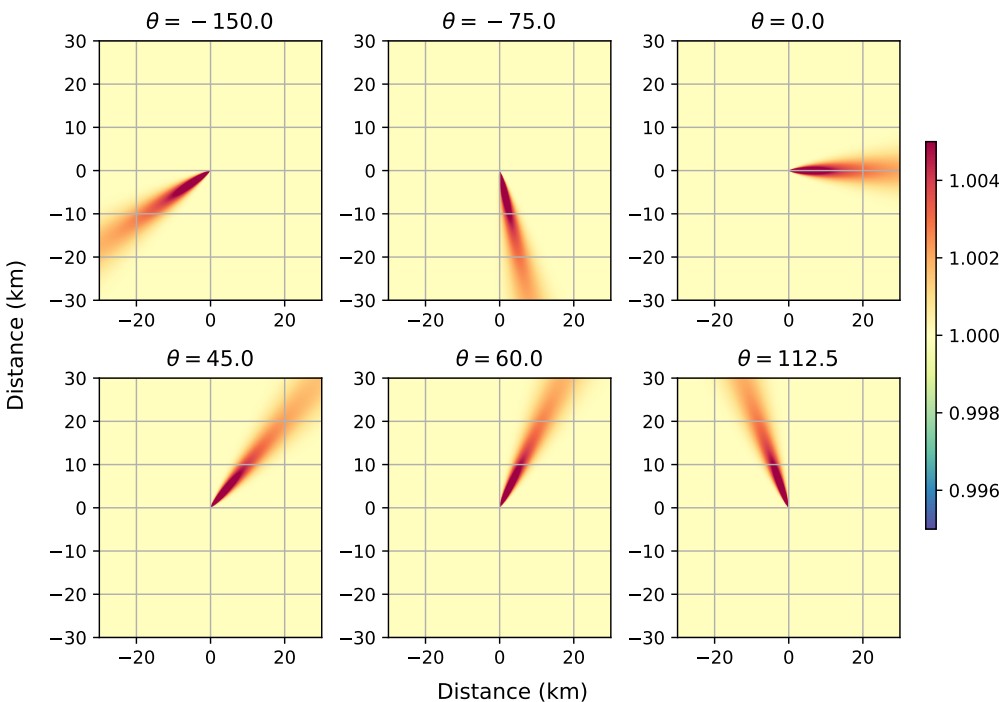

**Figure 3.** Simulated images of XCO$_2$ concentrations from a 13 Mt yr$^{-1}$ source relative to a constant 400 ppm background for six different wind directions. Angles are measured counter clockwise from the positive *x*-axis, or eastward. Panel labels identify the angle used for each panel.

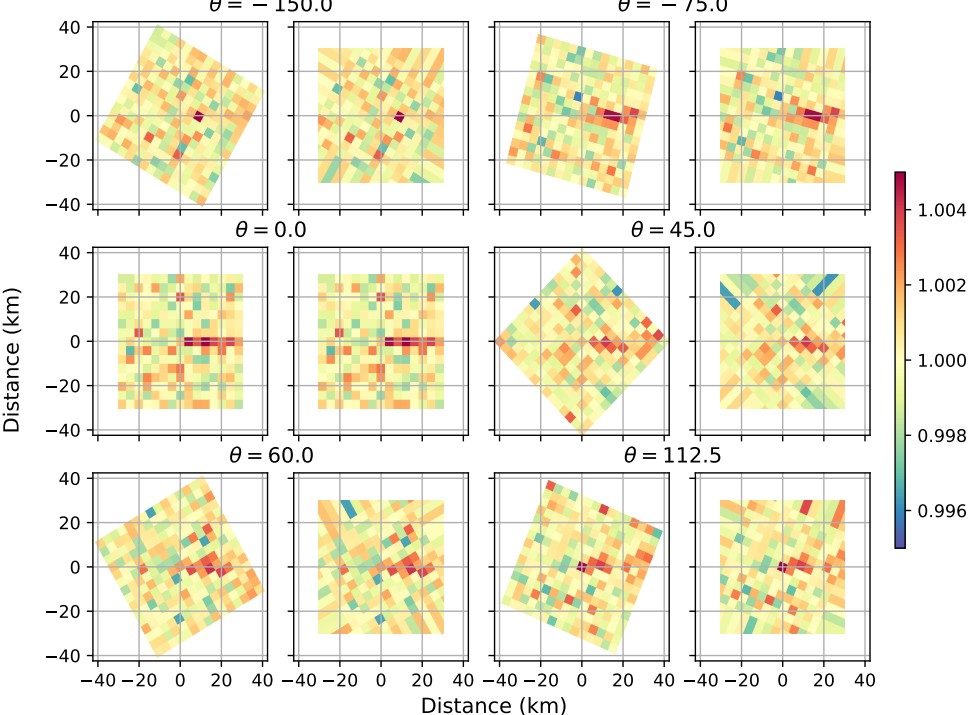

**Figure 4.** Simulated noisy XCO$_2$ observations relative to a 400 ppm background for $4 \times 4$ km$^2$ pixels after rotating to align the plumes and interpolating the rotated grids back onto a regular grid aligned with the plume. Columns 1 and 3 show the images at their 4 km $\times$ 4 km pixel resolution immediately after rotating to align the plumes. Columns 2 and 4 show the images after interpolating onto a higher resolution (50 m) regular grid using a nearest neighbour method.

To more accurately estimate the source emission rates, the emission sensitivity function $\alpha$ (Equation (3)) must be computed in the same way as the simulated observations. It is computed for each wind direction, averaged to the pixel size, rotated back to $0°$, interpolated onto the regular grid, and the samples are averaged to determine the final value.

### 2.7. US Power Plant Emissions Data

Nassar et al. [4] showed that there were at most a few good overpasses for each power plant in 28 months of OCO-2 data. Each good overpass results in a nearly instantaneous $CO_2$ emission estimate with some associated uncertainty. Due to sub-daily and seasonal variation in the power plant emissions, this emission estimate cannot simply be extrapolated directly to an annual value. However, if we understand the diurnal cycle and seasonality of power plant $CO_2$ emissions, we can estimate the number of revisits needed to achieve a desired precision for an annual emission estimate for a given precision on each individual estimate.

The 50 highest emitting US power plants as reported by the US EPA air markets program (https://ampd.epa.gov/ampd/) were studied. The 2016–2018 daily emission data and 2018 hourly data from the US EPA (ftp://newftp.epa.gov/DMDnLoad/emissions/) are used to study variability in emissions. Emissions were determined by using a continuous emission monitoring system (CEMS) at these facilities. Data were converted from short tons to metric kilotonnes (kt) of $CO_2$ per day and per hour.

## 3. Results

### 3.1. Pixel Size

We investigate the ability to accurately and precisely estimate power plant $CO_2$ emissions with varying pixel sizes, considering noise in the observations and biases in the observations or methodology. First, four sizes of square pixels are compared. Then, we compared rectangular pixels to square pixels of nearly equal area. The accuracy and precision of $CO_2$ emission estimates are quantified by computing the mean and standard deviation of emission estimates from an ensemble of 30 simulations.

Square pixels with side lengths 2, 4, 7, and 10 km were used. The simulation was run 30 times, with only the random noise varying between members of the ensemble. The ensemble standard deviation was used to quantify the precision of each estimate, and the ensemble mean allows for a more robust estimate of the accuracy in each case than any individual member.

### 3.1.1. Square Pixels with Biases

The $XCO_2$ observations relative to the constant 400 ppm background for the baseline experiment with no external biases are shown in Figure 1. The quantitative emission estimates are shown in Figure 5, which shows little difference between the accuracy or precision of the emission estimates for different pixel sizes. The high accuracy emission estimates resulting from all pixel sizes were a result of including no biases in the simulation. The emission estimates from the $2 \times 2$, $4 \times 4$, and $7 \times 7$ km$^2$ pixels all had approximately 2.5 Mt yr$^{-1}$ precision for all emission rates. This was equivalent to 42%, 19%, and 10% precision for the 6, 13, and 25 Mt yr$^{-1}$ sources, respectively. The $10 \times 10$ km$^2$ km pixels have 3 Mt yr$^{-1}$ precision, or 50%, 23%, and 12% precision for each source emission rate, respectively.

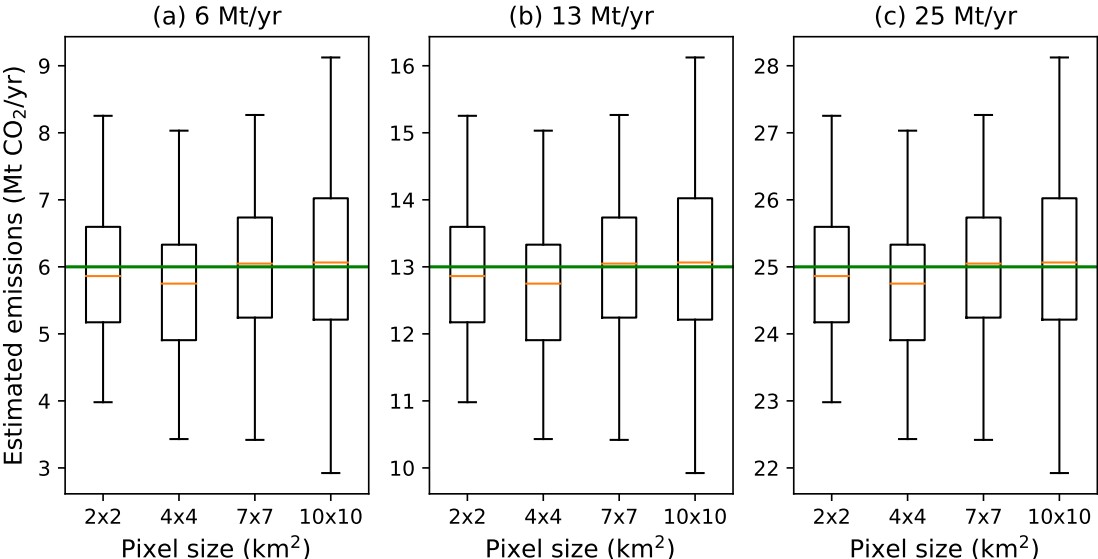

**Figure 5.** Emission rate estimates for all 12 combinations of pixel sizes and emission rates for an ensemble of 30 simulations. The green line is the true simulated emission rate, the orange line in each box is the median emission estimate, the black boxes show the range between the lower and upper quartiles (the interquartile range, IQR), and the whiskers (capped black lines) extend from the smallest datum within 1.5 times the IQR below the box to the largest datum within 1.5 times the IQR above the box. Outliers outside this range are shown as open circles. (**a**) Emission rate estimates for a 6 Mt yr$^{-1}$ source. (**b**) Emission rate estimates for a 12 Mt yr$^{-1}$ source. (**c**) Emission rate estimates for a 25 Mt yr$^{-1}$ source.

The consistent precision across the pixel sizes is expected for this baseline experiment with square pixels. The SNR and maximum plume enhancements are both inversely proportional to the pixel linear side length. Since no external biases are included, the ability to estimate the emissions is primarily controlled by the ratio of the SNR and the maximum enhancements in the plume. Thus, we expect similar results across the pixel sizes. The slightly worse precision of the 10 km pixel illustrates that with decreased relative enhancements the emission estimation method is more sensitive to slight changes in the background noise.

The ensemble is also run with a combination of observational biases and a methodological bias for a more realistic scenario. The observational biases are a topography related bias (max XCO$_2$ anomaly 1.2 ppm), a footprint bias (max XCO$_2$ anomaly 0.4 ppm), and an albedo bias (max XCO$_2$ anomaly 0.4 ppm). The methodological error is a mismatched atmospheric stability parameter between the modelled plume ($a = 130.0$ m) and the simulated data ($a_{sim} = 156.0$ m). Biases are described in detail in the supplementary materials. The topography bias was related to the gradient of the surface elevation, similar to the bias that was corrected in OCO-2 V9 data [9]. The simulated data with this combination of biases is shown in Figure 6, and the emission estimates are shown in Figure 7.

Figure 7 shows large systematic errors due to the included biases when compared to the baseline simulation (Figure 5). The 2, 4, and 10 km pixel sizes have similar accuracy, while the 7 km pixel size has noticeable larger systematic biases. For the 6 Mt yr$^{-1}$ source, the 2, 4, and 10 km pixels systematically underestimate the emissions by approximately 3 Mt yr$^{-1}$ (50%), while the 7 km pixel systematically underestimates the emissions by approximately 4.5 Mt yr$^{-1}$ (75%). For larger sources the absolute bias is larger, but the percent bias decreases. For the 25 Mt yr$^{-1}$ source, the systematic biases are approximately 16% for the 2, 4, and 10 km pixels, and 22% for the 7 km pixels. The precision of the estimates is similar to the baseline experiment (Figure 5). However, with the systematic errors induced by the biases, the outside of the range of estimates for the 7 × 7 km$^2$ pixel extends into negative emission rates.

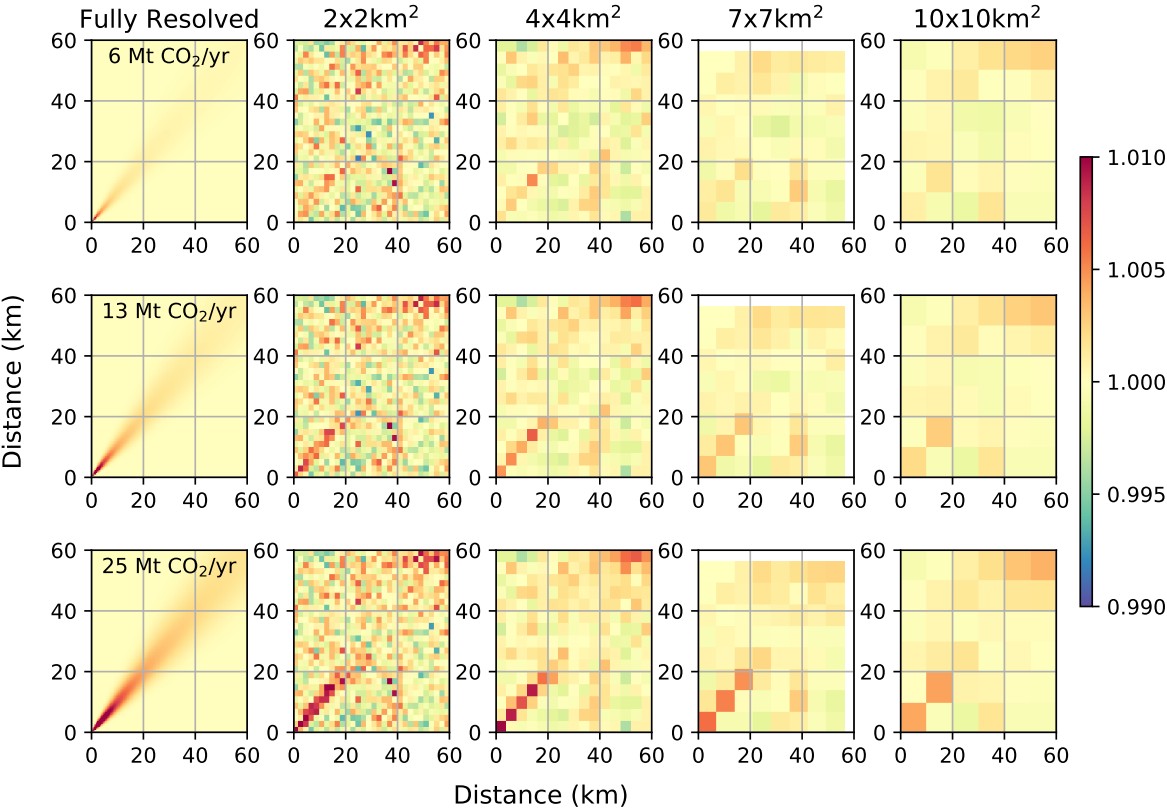

**Figure 6.** Simulated and model XCO$_2$ observations relative to a constant 400 ppm background with topography, footprint, albedo, and stability class biases. Panels are as in Figure 1.

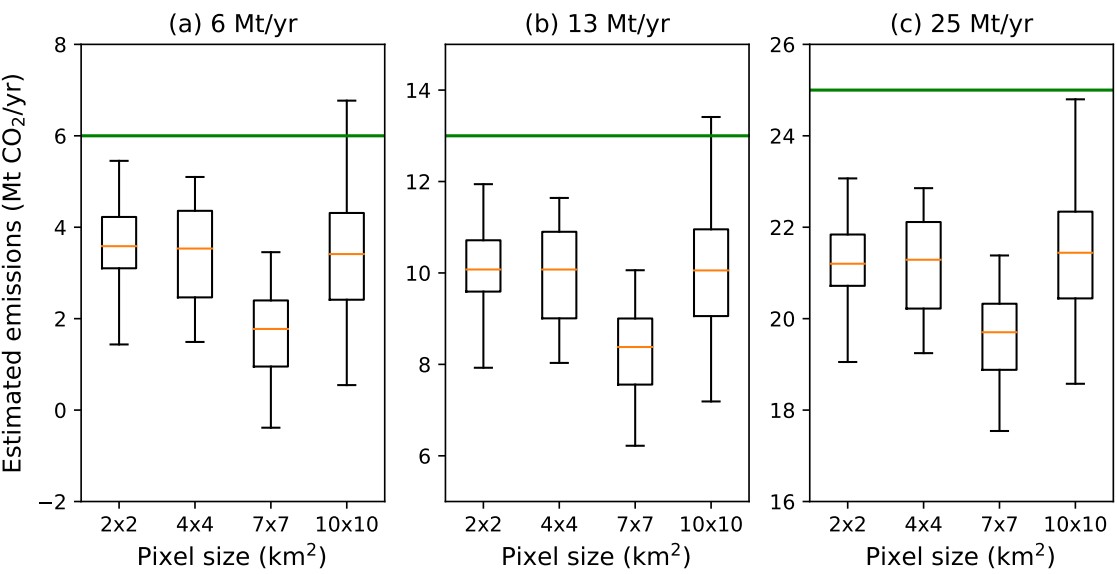

**Figure 7.** Emission rate estimates for all 12 combinations of pixel sizes and emission rates for an ensemble of 30 simulations as in Figure 5, with topography, footprint, albedo, and stability class biases.

We consider a 10° difference in wind direction between the simulated data and the model plume used to estimate source emissions. When including only a wind direction error, the accuracy of each pixel size is either constant or is better for larger pixels. Figure 8 shows the ensemble emission estimates when a 35° wind direction is assumed for the emission estimates and 45° is used to simulate the data. For all source rates, the smaller 2 and 4 km pixels showed an advantage in

precision over the larger 7 and 10 km pixels. However, the accuracy improved for larger pixels. For the 6 Mt yr$^{-1}$ source, the 2 and 4 km pixels underestimated the emissions by approximately 2.5 Mt yr$^{-1}$ (42%), while the 7 km pixel underestimates the emission by 1.5 Mt yr$^{-1}$ (25%), and the 10 km pixel underestimates by 2 Mt yr$^{-1}$(33%). For the 25 Mt yr$^{-1}$ sources, the 2 and 4 km pixels had similar accuracy, underestimating the emissions by 8 Mt yr$^{-1}$ (32%). The 7 and 10 km pixels both underestimated emissions by 5.5 Mt yr$^{-1}$ (22%). The percent errors were similar for the 13 Mt yr$^{-1}$ source, showing a 10% improvement in accuracy for larger pixels. However, smaller pixels would enable a much better manual adjustment of the wind direction to match the plume direction, as in Nassar et al. [4]. With the large pixels it would be difficult to adjust the direction to match the observations because the plume enhancement is relatively weak and the plume shape is poorly defined.

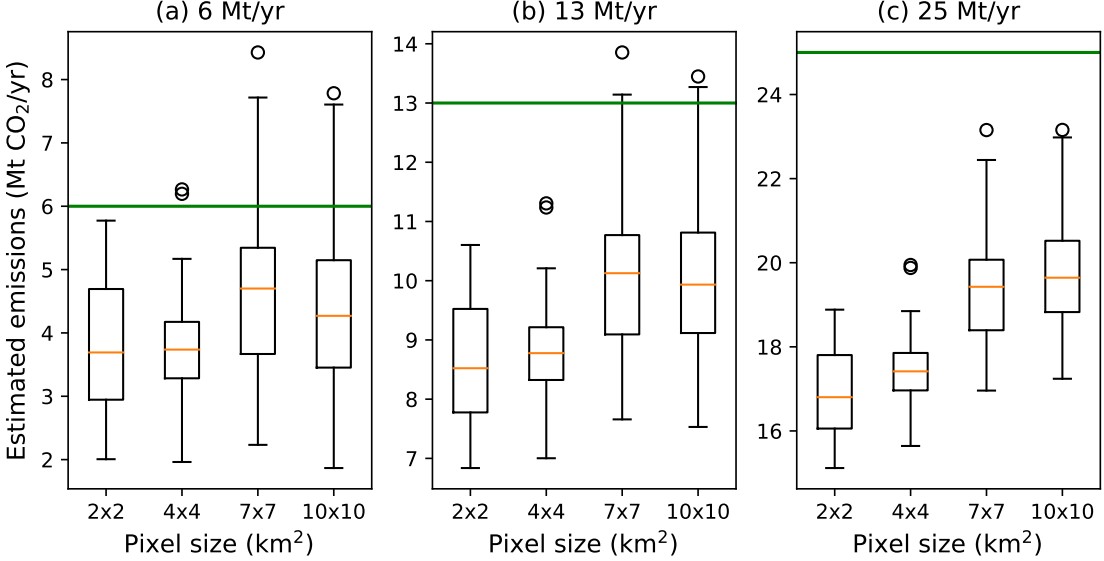

**Figure 8.** Emission rate estimates for all 12 combinations of pixel sizes and emission rates for an ensemble of 30 simulations as in Figure 5, with a simulated wind at 45° and a modelled direction of 35°.

Results for remaining biases are shown in Figures S7–S15. The emission estimates for the footprint bias (Figure S8), both swath masks representing a narrow swath width (Figures S11 and S12), and the stability class bias (Figure S13) show a precision advantage for smaller pixels. For the topography bias (Figure S7), footprint bias (Figure S8), and interfering source (Figures S14 and S15) cases the accuracy is best for 2 km pixels and decreases for larger pixels. The surface albedo bias (Figure S10) increases the uncertainty in all pixel sizes similarly, hence there is no clearly preferred pixel size. The case with lost observations due to clouds (Figure S9) shows the 2 km pixels had the most trustworthy emission estimates, and otherwise varied results. The 4 km pixels had both the lowest accuracy and the highest precision as measured by the whisker extent. However, 4 and 7 km pixels were susceptible to outliers, and 10 km pixels had significantly lower precision than the other pixel sizes.

Figure S14, showing the simulated XCO$_2$ observations with a weak interfering source, illustrates one of the additional difficulties of large pixel sizes. This case includes a weak (5 Mt yr$^{-1}$) source near the primary source in the simulated XCO$_2$ observations, but this second source is not included when computing the enhancement sensitivity function $\alpha$. This represents an unknown contaminating source in the observations. The plume from this secondary source is only clear in the observations from the $2 \times 2$ km$^2$ pixel, which would allow the source to be accounted for, whereas the source would remain unknown for the larger pixels, leading to significantly biased estimates.

### 3.1.2. Rectangular Pixels

Rectangular pixels ($2.3 \times 7$ km$^2$) are compared to square pixels of approximately equal area ($4 \times 4$ km$^2$), with no other biases applied. Over the ensemble of 30 simulations, rectangular pixels performed consistently worse than square pixels. Figure 2 shows the simulated observations, and Figure 9 shows the emission estimates for all three source emission rates and three pixel sizes and orientations. While the accuracy of each pixel is similar, the precision of the rectangular pixels is worse. Moreover, estimates from pixels with the long edge oriented perpendicular to the plume were significantly less precise than estimates from pixels with the long edge oriented parallel to the plume. One contributing factor can be found by studying Figure 2. When the pixels have their long edge across the plume, they average the areas of very high enhancements near the source with areas outside the plume, resulting in relatively low maximum enhancements. In contrast, pixels with their short edge perpendicular to the wind mostly average along the wind direction, which decays much slower, and so they maintain higher maximum enhancement values, and allow for more precise emission estimation. In this case, the impact on emission estimates is similar to that of having a narrow field of view (Figure S11).

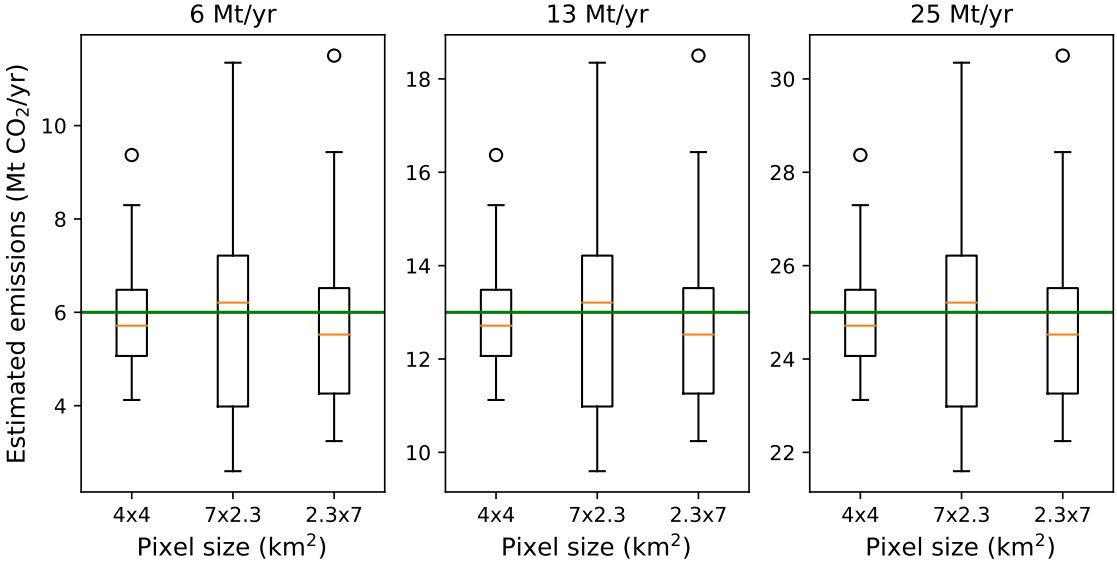

**Figure 9.** Emission estimates from an ensemble of 30 simulations for square pixels ($4 \times 4$ km$^2$), and rectangular pixels with the long side oriented both perpendicular to ($7 \times 2.3$ km$^2$) and along ($2.3 \times 7$ km$^2$) the wind, as in Figure 5.

Due to the absence of imposed biases, the difference between the median estimated and true emissions is negligible. However, the rectangular pixels have significantly worse precision than the square pixels. The square pixels have approximately 2 Mt yr$^{-1}$ precision in emission estimates, while the rectangular pixels oriented across the plume have only 4 Mt yr$^{-1}$ precision, and the rectangular pixels oriented along the plume pixels have approximately 3 Mt yr$^{-1}$ precision. This corresponds to up to a two times more precision estimate with square pixels than rectangular pixels. These results are consistent with those from Polonsky et al. [10]. They found that the number of images required to reduce the error in the estimated rate of CO$_2$ emissions to below 1 Mt yr$^{-1}$ was much greater if the pixels were smeared across the wind direction than along the wind direction.

*3.2. Averaging Multiple Images*

The average of six individual images with wind directions $-150°$, $-75°$, $0°$, $45°$, $60°$, and $112.5°$ is shown in Figure 10. An ensemble of 30 simulations was run to estimate the accuracy and precision of this method. The estimated emissions across the ensemble are $13.143 \pm 0.566$ Mt yr$^{-1}$ (4.35% precision), where the true simulated emissions were 13.0 Mt yr$^{-1}$. Observations from individual images with no averaging led to an emission estimate of $13.137 \pm 1.886$ Mt yr$^{-1}$. The estimate from the averaged images represents better than a three times improvement in the precision of the emission estimate. This improvement was better than the statistical increase of $\sqrt{6} \approx 2.5$ that we would expect having six times as many observations. Figure 10 shows the averaged images compared to the theoretical XCO$_2$ enhancements using the estimated emission rate. We see the plume is significantly more defined in comparison to the background noise. In Figure 4 the plume was still distinguishable, but after averaging the observations the background noise nearly disappears and the plume is very prominent.

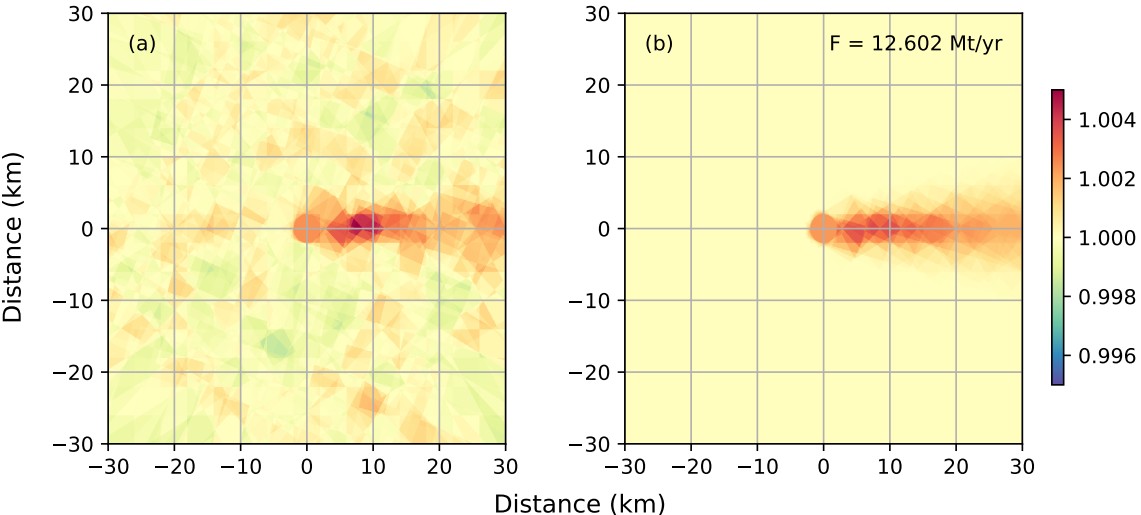

**Figure 10.** XCO$_2$ observations relative to a 400 ppm background for one instance of the 30 member ensemble. (**a**) Average observations from 6 simulated images at varying wind directions for a 13 Mt yr$^{-1}$ source. (**b**) Fitted relative XCO$_2$ enhancements from the least squares fit. The label identifies that the estimated emission rate was 12.602 Mt CO$_2$ yr$^{-1}$ for this individual ensemble member.

The advantages of this method are further illustrated by considering a weaker 6 Mt yr$^{-1}$ source. The individual images are shown in Figure 11. For such a weak source, the plume is nearly indistinguishable from the background noise. After averaging the six images, the plume clearly stands out from the background noise (Figure 12). While not as obvious as the 13 Mt yr$^{-1}$ source, the plume was still discernible. Across an ensemble of 30 simulations, the emission rate was estimated as $6.2055 \pm 0.6918$ Mt yr$^{-1}$, or 11.5% precision. Using individual images with no averaging, we estimated the emissions to be $6.1646 \pm 1.7639$ Mt yr$^{-1}$, or 29.4% precision. By averaging images, we again achieve a nearly three times improvement in precision.

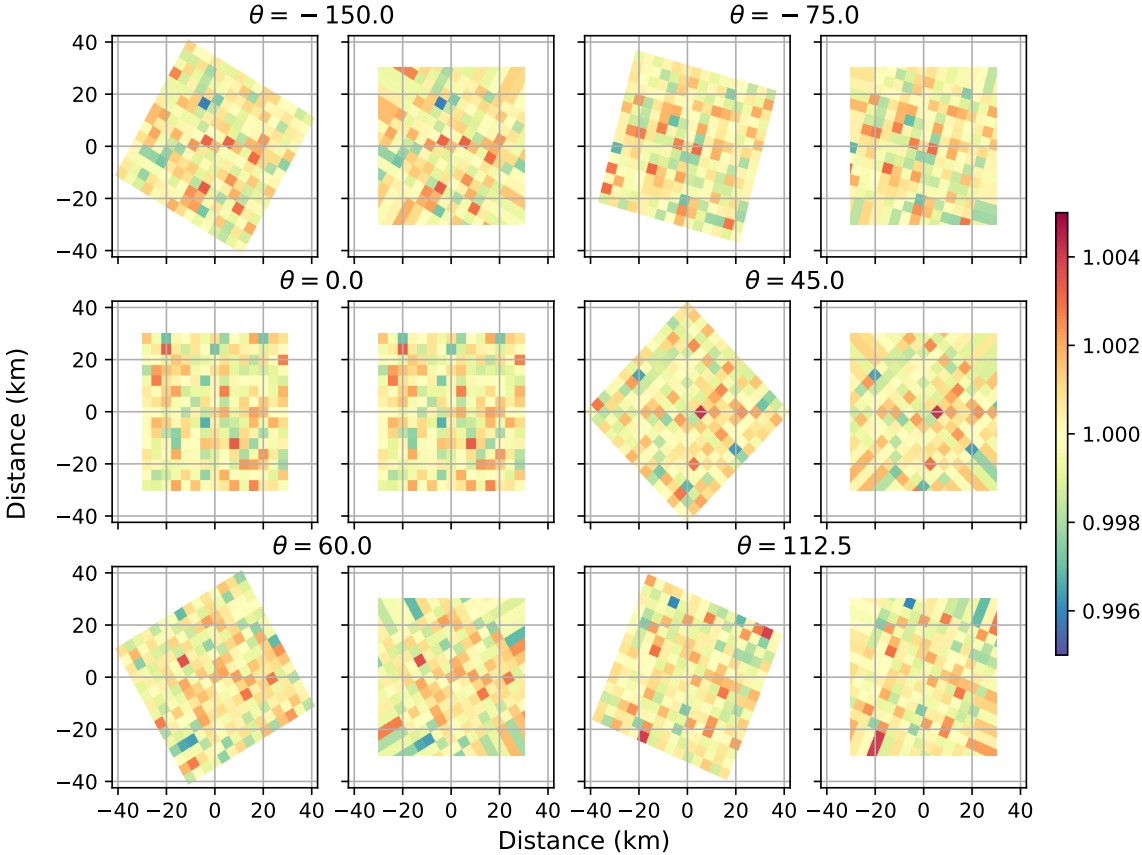

**Figure 11.** Simulated images of $XCO_2$ relative to a 400 ppm background as in Figure 4, for a 6 Mt $yr^{-1}$ source.

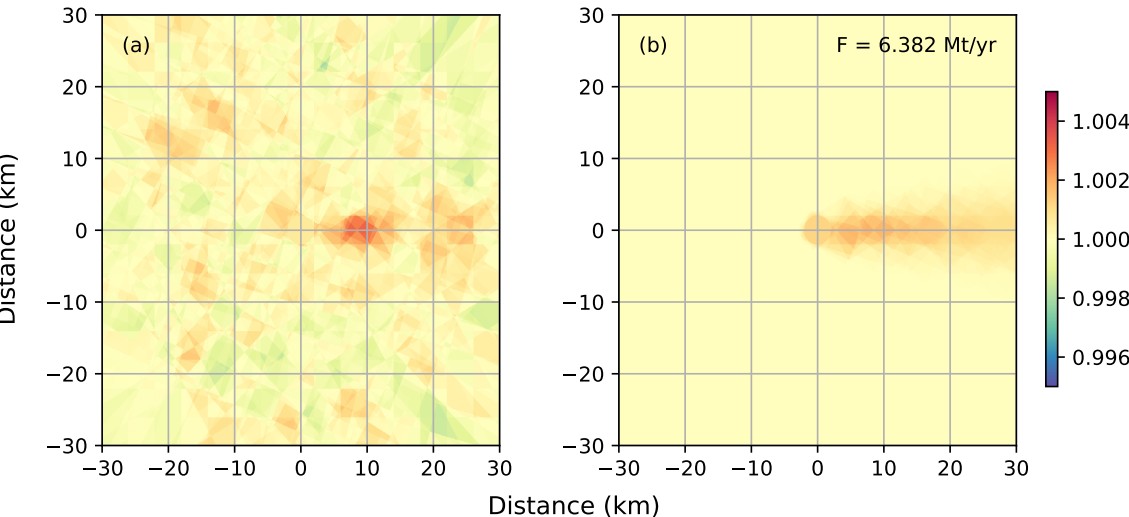

**Figure 12.** $XCO_2$ observations relative to a 400 ppm background for one instance of the 30 member ensemble for a 6 Mt $yr^{-1}$ source. (**a**) Average observations from 6 simulated images at varying wind directions. (**b**) Fitted relative $XCO_2$ enhancements from the least squares fit. The label identifies that the estimated emission rate was 6.382 Mt $yr^{-1}$ for this ensemble member.

An ensemble of 10 simulations was run with 16 wind directions, from which we estimate the emissions to be $5.923 \pm 0.407$ Mt yr$^{-1}$, or 6.78% precision. The averaged estimate represents more than a two times improvement in accuracy and more than a four times improvement in precision when compared to estimates from individual images.

### 3.3. Revisit Rate

It is well known that anthropogenic $CO_2$ emissions exhibit seasonal and diurnal variability [11,12], with the day-night difference for U.S. power plants previously estimated to be $\sim$9% [13]. In this section we study the diurnal and seasonal variability using 2016–2018 daily emissions data and 2018 hourly emissions data for the 50 largest US power plants (based on 2017 reported emissions). We compute the mean weekly and hourly emission factors, and use these factors to remove the seasonal and daily cycles to achieve a true estimate of the variability in US power plant emissions.

Mean weekly and hourly cycles were computed and then removed from the data. The data for each plant was normalized to its mean emissions for the time period studied, resulting in the dimensionless daily and hourly relative emissions. The relative emissions for all 50 plants are aggregated and binned by week of year and by hour of day. The last eight days (nine days for the leap year 2016) of the year were grouped together in the last weekly bin of the year, allowing us to compute 52 weekly emission factors. The mean weekly and hourly relative emissions were computed, and then normalized so that the sum of the weekly factors was 52 and the sum of the hourly factors was 24. This last normalization step ensures the mean emissions remain unchanged when removing the weekly and hourly cycles.

Figure 13 shows the weekly emission scale factors, with error bars showing the standard deviation in the binned weekly emissions.

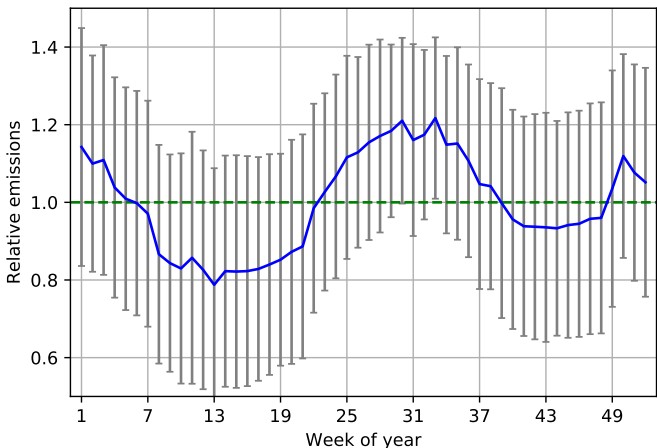

**Figure 13.** Weekly emission scale factors for 50 highest emitting US power plants, data from 2016–2018. Error bars show the standard deviation in the binned weekly relative emissions.

Deseasonalized daily emissions were computed from the weekly scale factors by dividing the emissions for every day in each week by the scale factor for the week, with no dependence on the day of the week. We compared the residuals, defined as the difference between the deseasonalized daily emissions and the mean emissions, normalized by the mean emissions, to a standard Normal distribution. Figure 14 shows the quantiles of the observed data vs. quantiles of a theoretical Normal distribution. Other than the lower tail of the relative emissions, the normal approximation fits the residuals very well. Moreover, removing the seasonal cycle significantly improved the fit. This was further evidence that we have both found and removed the seasonal cycle within the data. The daily emissions for all 50 power plants for 2016–2018 are shown in Figure S16. The daily emissions showed a strong seasonal cycle. After removing the seasonal cycle by dividing by the weekly scale factors,

no periodicity is evident in the mean residual emission rates. Finally, we can estimate the standard deviation of daily power plant emissions. Computing the daily standard deviation of the residuals, we find $s_d = 0.286784$.

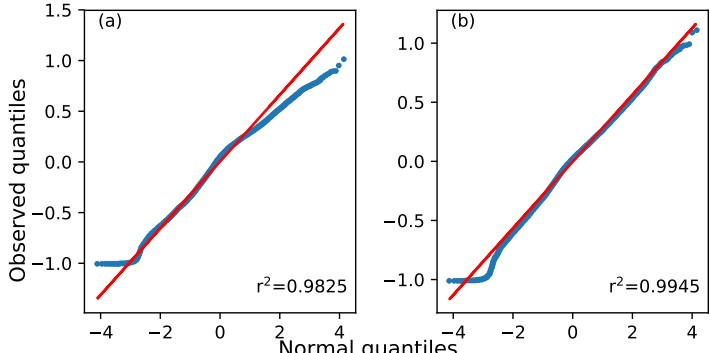

**Figure 14.** Quantile-quantile plot of relative daily emission data with squared correlation coefficient $r^2$. Red line shows the expected distribution if the residuals are perfectly normally distributed. (**a**) Relative emissions with no weekly factor adjustment. (**b**) Relative emissions with weekly cycle removed.

We similarly investigate the hourly variability in power plant emissions. Figure 15 shows the hourly scale factors derived from all 50 sources. Hourly scale factors were derived from hourly emissions data after using the weekly scale factors shown in Figure 13 to remove any seasonal dependence. In this way, the hourly scale factors exclude any seasonal component. However, we note that in general, daily energy demand will have a different distribution throughout the day in different seasons. For example, the length of the overnight minimum in power plant emissions might depend on the number of daylight hours, or the shape of the midday peak may depend on air conditioning demand. These factors are excluded, and any seasonality is relegated to the weekly scale factors.

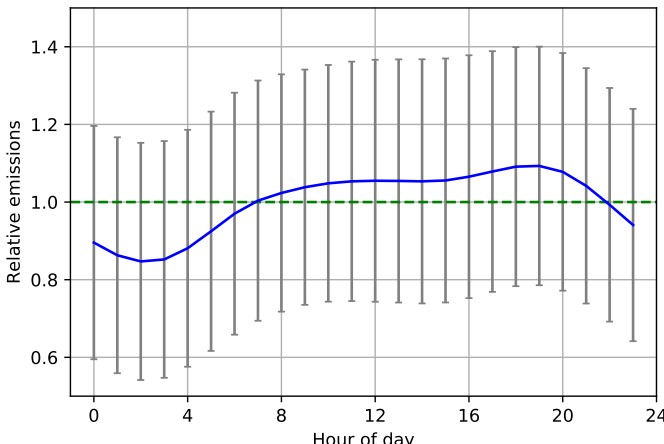

**Figure 15.** Hourly scale factors for 50 highest emitting US power plants, using 2018 data. Error bars show the standard deviation in the binned hourly relative emissions.

Figure 16 shows the hourly emissions from all 50 sources for all of 2018. Notice the strong seasonal cycle in the top panel. After removing the weekly and hourly cycles, the residuals in the bottom panel appear much more randomly distributed about zero.

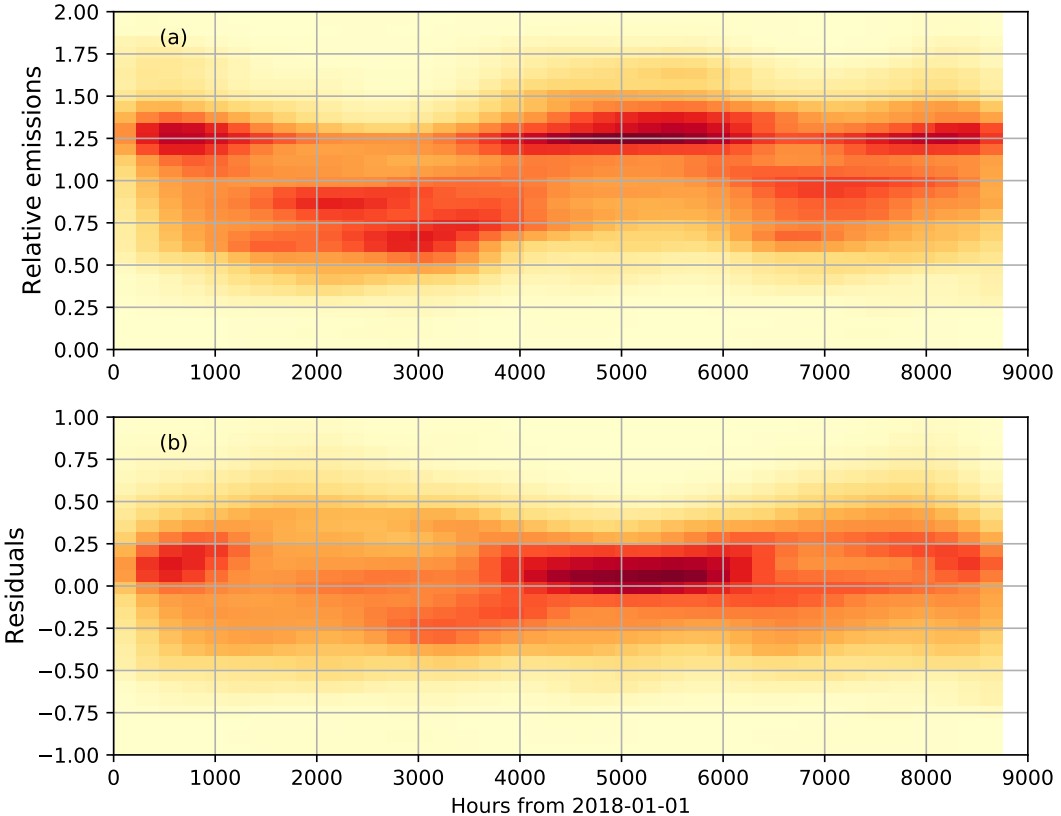

**Figure 16.** 2D histogram of relative hourly power plant emissions for 2018. (**a**) Relative emission data from all 50 plants, with no adjustments. (**b**) Difference between emissions with weekly and hourly factors removed and mean emissions, normalized by the mean emissions.

In the same way as daily emissions, we compare the hourly emissions to a Normal distribution in Figure 17. We note in particular that the fit of a Normal distribution is not very good before removing hourly and weekly cycles. The quantiles disagree noticeably across most of the Normal quantiles in addition to in the tails. After adjusting the emissions by the hourly and weekly scale factors, the fit improves significantly. The second panel shows that the Normal approximation is good for this dataset. Computing the standard deviation of the residuals, we find the hourly standard deviation is $s_h = 0.310096$.

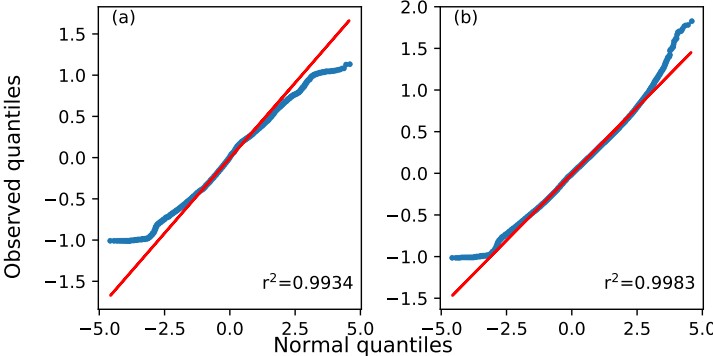

**Figure 17.** Quantile-quantile plot of relative hourly emission data with squared correlation coefficient $r^2$. Red line shows the expected distribution if the residuals are perfectly normally distributed. (**a**) Relative emissions with no hourly factor adjustment. (**b**) Relative emissions with hourly and weekly cycles removed.

We used a statistical approach based on a theoretical Normal distribution to determine the number of revisits needed to constrain annual emissions from an individual power plant. Treating the data as Normally distributed, we compute the number of samples required to achieve a desired width of 95% confidence interval. If $N$ is the number of revisits, $\bar{y}$ is the mean emission estimate and $\sigma$ is the uncertainty in each estimate, then the 95% confidence interval is $\bar{y} \pm \bar{y}\delta$, where

$$\delta = 1.96 \frac{\sigma}{\sqrt{N}}. \tag{4}$$

Hence, for a desired precision $\delta$, the number of required revisits is

$$N = \left(1.96 \frac{\sigma}{\delta}\right)^2. \tag{5}$$

We compute the total uncertainty $\sigma$ as a combination of the underlying standard deviation of the power plant emissions, and the uncertainty in each estimate, treating each estimate as an independent sample. The standard deviation in the daily relative emissions as computed from the 50 sources is $s_d = 0.286784$, and the standard deviation in the hourly relative emissions is 0.3100965. Nassar et al. [4] estimated US power plant emissions to within 1%, 4%, and 17% of EPA reported daily values. Their method absorbs any daily variability in power plant emissions in their sample uncertainty. From this small sample size, it was not clear what the precision and accuracy limits for a single estimate should be. Furthermore, preliminary results with OCO-2 version 8 and 9 data and the same power plants show improvements to both the precision and accuracy of emission estimates. Since we have now removed the daily variability, we may assume a 10% uncertainty on each individual emission estimate ($\varepsilon_{est} = 0.10$), which was very likely achievable with minor improvements to the data and methods. The total uncertainty is therefore

$$\sigma_h = \sqrt{s_h^2 + \varepsilon_{est}^2} \approx 0.3258. \tag{6}$$

To constrain annual emissions within at least 15%, we need $N = 19$ overpasses. To constrain annual emissions within 20% would need only 11 overpasses, but to constrain annual emissions within 10% would need 41 overpasses. If the individual estimate uncertainty can be reduced to 5%, we still require 38 overpasses.

Alternatively, we may absorb any daily emission variation into the sample uncertainty as in Nassar et al. [4] and used the standard deviation of daily emissions to compute the required number of overpasses. Increasing the sample uncertainty to $\varepsilon_{est} = 0.15$ to absorb the daily emission variation, we compute $\sigma_d = \sqrt{s_d^2 + \varepsilon_{est}^2}$. Following Equation (5), we require 18 overpasses to constrain annual emissions within 15%.

Velazco et al. [13] attempted to address the question of uncertainties for annual emission estimates from one CarbonSat satellite or a constellation, using hourly data from 187 U.S. power plants ($\geq 5$ Mt $CO_2$ yr$^{-1}$). They found that with one satellite, the accuracy on the annual estimate (or sampling error) was ~4.9% or better for 50% of the power plants and ~12.4% or better for 90% of the power plants. This assumed a 500 km wide swath, resulting in about 50 overpasses of a midlatitude power plant. The fraction of these overpasses that would be sufficiently cloud-free varied significantly according to local climate conditions, but clear-sky probability was expected to be ~30–70% for the majority of the continental U.S. based on MODIS. These numbers suggested that ~15–35 clear-sky overpasses (mean of 25) would give annual emissions within ~12.4% for 90% of power plants. The number of overpasses required seemed consistent with our findings which suggest that to get annual emissions with a precision of 12.4% at the 95% confidence interval, ~25 overpasses with an individual daily emission uncertainty of 10% would be required.

## 4. Discussion

This collection of simulations suggests that the precision and accuracy of emission estimates are quite similar for pixel sizes in the range of 2–10 km, when noise is included in the observations but biases are not. In a more realistic scenario, where small biases are present in the data and the method, smaller pixels have an advantage. The primary exception occurs in the isolated case of a wind direction bias, where the largest pixels gave the most accurate emission estimate, but this methodological bias is detectable and correctable with smaller pixels.

In general, the biases that are present are the most obvious for the smallest pixels, and become more subtle for larger pixels. Therefore, the smallest pixels have an advantage beyond the quantitative emission estimates since they would best enable some of the biases, particularly the methodological errors, to be corrected before reporting final emission estimates. In such cases the 2 km pixels would have emission estimates as shown in Figure 5, whereas the larger pixels will have significantly worse estimates, depending on the biases present.

It should be noted however, that our conclusion that smaller pixels are better does not necessarily extend to all spatial scales. Varon et al. [14] showed that sub-kilometre scale plume structure is dominated by eddies. We have not investigated pixels finer than 2 km due to the expectation that smaller pixels would likely require relaxing precision requirements. Due to the complexity and uncertainties in modeling plume behaviour when dominated by eddies, for $CO_2$ emission quantification, an optimum pixel size may be the ∼1–2 km scale, below which there is likely more benefit to greater precision than even smaller pixels. Broquet et al. [15] investigated $CO_2$ imaging with pixels in the range of 2–10 km for quantifying urban emissions and found best results were achieved with pixels of 4 km or smaller, which suggests that pixel size requirements for these two anthropogenic applications are roughly consistent. From Figure 9, it is clear that square pixels perform better than rectangular pixels of equal area. The emission estimates using square pixels have significantly less uncertainty than those using rectangular pixels, especially when they are aligned across the plume, a finding that is consistent with the power plant emission estimation studies in Polonsky et al. [10]. In real data with rectangular pixels, we would expect a mixture of observations with the long edge varying from being aligned along to across the plume, and any angle between, due to variation in the wind direction. Thus, the worst case scenario of the long edge oriented across the plume must be considered when predicting the performance of any instrument with rectangular pixels.

We have shown that averaging multiple individual images results in significantly improved emission estimates when compared to using the individual images directly. We found that by averaging six images of a medium-large sized (13 Mt yr$^{-1}$) source we estimated the emissions to be 13.143 ± 0.566 Mt yr$^{-1}$, or a precision of 4.35%. This was better than a three times improvement on the individual estimates, which estimated the emissions to be 13.37 ± 1.886 Mt yr$^{-1}$, or 14.5% precision.

Moreover, we were able to discern a plume from a small source (6 Mt yr$^{-1}$) that we could not discern in the individual images. We estimated its emissions with a precision of 11.5% using six images, and 6.78% using 16 images. In cases where the data is available, this method is a significant step forward when compared to utilizing individual observations. However, it is unclear how this method may improve observations when being applied to data from an instrument with a narrow swath such as OCO-2.

We estimated that approximately 19 individual overpasses are needed to constrain annual emissions within 15%, and we derived hourly and weekly emission scale factors. These scale factors could be used whenever estimating annual emissions from an instantaneous emission estimate. For instance, OCO-2 observations over the US always occur in the afternoon, when emissions are above the daily mean. If the daily cycle is not corrected for, the annual emission estimates will be systematically over-estimated.

Achieving the required number of overpasses with a single Low Earth Orbit (LEO) satellite now seems unlikely in the near future. By the end of CarbonSat's phase A/B1, the design had evolved to a 185–240 km swath [16], similar to plans for the Copernicus $CO_2$ monitoring mission currently under

consideration by the European Commission in partnership with ESA. Furthermore, past theoretical studies expected that a 4 km$^2$ pixel would deliver about 23% cloud-free observations [17,18] but real OCO-2 statistics and a smaller pixel size ($\leq$3 km$^2$) suggest only 7–12% of observations per month pass all data screening filters (clouds and others) [6]. With about half the swath width and one fifth of the clear-sky observations (7–12% instead of 30–70%) from Velazco et al. [13], rather than 50 overpasses for a typical midlatitude site, a single satellite would likely obtain $\sim$5 clear-sky overpasses/year. A constellation of multiple wide-swath LEO missions is of course one way to obtain the required revisit rate, however geostationary (GEO) observations are another approach, with the potential for sub-daily revisit rates over land from $\sim$50°N–50°S [10,19] and limited glint observations. Quasi-geostationary observations of high latitude regions can be obtained using from a highly elliptical orbit (HEO) [20,21]. Including GEO and HEO in the constellation, along with LEO is the longterm vision for CO$_2$ monitoring for Europe [2] and suggested for the global architecture by CEOS [22]. Including GEO and HEO would likely lower the total number of satellites required since these vantage points better facilitate frequent revisits for key regions and flexible, cloud-informed pointing strategies can improve efficiency by timing observations for cloud avoidance [21]. Regardless of orbit, passive measurements using reflected sunlight would still lack night time data. The number of overpasses required for annual emission estimates also suggests that a short-duration campaign approach as with aircraft [23] will be severely limited in providing annual emission estimates on its own; however, aircraft in situ measurements could be complementary to a satellite-based approach by providing night time data for better overall daily emission estimates of showing consistency with daytime satellite-based estimates.

By combining the new image averaging method and our estimates of revisit rates, we show that even for relatively weak sources ($\sim$6 Mt yr$^{-1}$), the overall approach is actually limited by the variability in the power plant emissions, not the uncertainty in daily emission estimates. If we were able to sample daily emissions exactly ($\varepsilon_{est} = 0$), we would still require 16 such samples to constrain annual emissions within 15%. We have also shown that with 16 observations, we can estimate the emissions from a 6 Mt yr$^{-1}$ source within 6.78%. Therefore, by averaging a collection of images, the limiting factor in estimating annual emissions is the power plant variability itself, instead of the uncertainty in the instantaneous estimates, even for sources we can not see in the observations from single images.

## 5. Conclusions

In this study, we investigated the impact of pixel size and shape on the accuracy and precision of power plant CO$_2$ emission estimates, presented and evaluated a method to average multiple images of CO$_2$ enhancements, and derived a simple expression for the required number of overpasses to quantify annual emissions with a desired precision, accounting for both the variability in power plant emissions and the uncertainty in the estimates from each overpass.

By studying square pixels with side lengths in the range of 2–10 km, we showed that the smallest pixels in this range have an advantage. Moreover, it would be easier to correct for biases in the observations with smaller pixels, particularly an error in the wind direction, which is one of the largest sources of error when using real meteorological data. By comparing square 4 $\times$ 4 km$^2$ pixels to rectangular pixels of approximately equal area with a three to one aspect ratio (2.3 $\times$ 7 km$^2$), we determined that square pixels have up to twice the precision of rectangular pixels.

A method to average multiple images of CO$_2$ enhancements from the same source was presented and was shown to increase the accuracy and precision of emission rate estimates significantly. For a medium-large (13 Mt yr$^{-1}$) source, we improved emission estimates by more than three times. For a small (6 Mt yr$^{-1}$) source, using six observations led to an improvement of nearly three times in precision, and using 16 observations led to a more than four times improvement in precision and two times improvement in accuracy.

The variability in hourly and daily power plant emissions was studied using data from the 50 highest emitting US power plant (as reported in 2017), and we found significant variability at

both time scales. Therefore, individual emission rate measurements can not be directly converted to annual emissions. Accounting for realistic estimates of the uncertainty in each estimate of power plant emissions and the variability in power plant emissions, we derived an expression for the number of overpasses required to achieve a desired precision in annual emission estimates. We found that at least 18 overpasses are required to constrain annual emissions within 15%. Hourly and weekly scale factors were derived that could be used to account for mean daily and seasonal cycles when estimating annual power plant emissions.

This work builds upon our earlier work that quantified $CO_2$ emissions from individual power plants using OCO-2 observations [4], by conducting simulations to investigate important questions that can inform the design of future satellite missions intended to have a greater emphasis on anthropogenic emission quantification. By combining studies utilizing real observations and simulations to explore new methods and synthetic observations, we are improving our understanding of the strict requirements for space-based systems to provide observations with the necessary characteristics to monitor anthropogenic $CO_2$ emissions in a manner that would support international climate agreements.

**Supplementary Materials:** The following are available online at http://www.mdpi.com/2072-4292/11/13/1608/s1. Figure S1: Simulated topography, Figure S2: Simulated topography bias, Figure S3: Footprint bias, Figure S4: Cloud mask bias, Figure S5: Albedo bias, Figure S6: Swath mask biases, Figures S7–S15: Results for individual biases, Figure S16: Daily power plant emissions for 2016–2018. The data files containing derived weekly and hourly scale factors are available online at http://www.mdpi.com/2072-4292/11/13/1608/s2.

**Author Contributions:** Conceptualization, R.N.; data curation, T.H.; formal analysis, T.H.; investigation, T.H.; methodology, T.H. and R.N.; software, T.H.; visualization, T.H.; writing—original draft, T.H. and R.N.; writing—review and editing, T.H. and R.N.

**Funding:** This research received no external funding.

**Acknowledgments:** The authors would like to acknowledge the US EPA for providing open-access hourly and daily emissions data at the facility level. We thank Chris Sioris and Felix Vogel of Environment and Climate Change Canada for helpful discussions and suggestions related to this study.

**Conflicts of Interest:** The authors declare no conflict of interest.

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
