# Peer review of "Pixel Size and Revisit Rate Requirements for Monitoring Power Plant CO2 Emissions from Space"

_remotesensing, doi:10.3390/rs11131608_

Round 1

Reviewer 1 Report

The manuscript “Pixel size and revisit rate requirements for monitoring power plant CO2 emissions from space” by Hill and Nassar is an interesting and well-considered study into the optimal design of a potential satellite measurement system for assessing point sources of CO2 emissions. The study is thorough in its assessment of the effects of potential biases and satellite functions, and also investigates required revisitation rates for high-precision estimates of point source emissions.

The manuscript is very well-written, and on the whole, the authors' reasoning is very clear. The figures are clear and well chosen and the methods and models used within the manuscript are appropriate for such a study. The terminology is consistent and the chosen equations are clear and appropriate. In my opinion, only very few technical corrections are necessary.

I recommend publication of this manuscript subject to the following minor changes.

Figure 4 caption and page 7, lines 151-154: The language here is a little unclear and this 'regridding' should be explained slightly better. It took me a few attempts to understand exactly what the authors are saying, and what columns 2 & 4 of Figure 4 are showing. Also, I don't think the fact that these simulations are carried out on a 30km x 30km grid is mentioned in the text. You should also say how the rotated grid is interpolated onto the regular grid for corner areas that aren't covered by the rotated grid. Is the nearest value used?

Page 9, lines 208 - 217: Is there a methodological reason why the 7km x 7km pixel size would consistently produce larger biases than the other resolutions, or is it simply the case that it might be due to the specific noise and bias fields produced for these particular 30-member ensembles?

Section 3.1.2: Similarly, for this particular case, what is the reason that the rectangular pixels with the long edges running parallel with the plume produce lower accuracy than the square pixels? Looking at Figure 2 by eye, there doesn't seem to be much information lost, or 'smearing' of the plume, in this case.

Figure 12 caption: Tthe -> The

Page 15, line 314: Just an extra sentence here would be nice to explain how the deseasonalised daily emissions were produced. Are the weekly emission scale factors applied even for every day in each 7-day period or is there some form of interpolation?

Page 20, line 455: Your estimate of the power plant variability appears to come from averaging the 50 sites. Would the power plant variability, and associated uncertainty, change if you create 'bespoke' seasonal/daily cycles for different plants, or for different regions of the US with different climates, or, as you do state earlier, if using seasonally-varying hourly scale factors? Or is there not enough data to produce such information?

Reviewer 2 Report

 The manuscript Remote Sensing #529292 by Hill and Nassar is interesting, timely, well written and fully relevant for the journal.

 Here are just some small comments

 1) The parameter "a" usually called "the atmospheric stability parameter" (as done in the manuscript) is standard but a little missleading. The reviewer is suggesting to introduce also the name "half-width of the plame at x=x0". This is only indirectly related to atmospheric stability.
 In the manuscript, the authors give the value a=156.0 without unit. In fact this value is in meter and the authors should clarify this point.

 2) The word "whisker" is used once but not defined precisely. Shouldn't it be "pixel"?

 3) Reference 13. to Velazco is incomplete. Just a year (2011) is given.

 4) The reviewer has not been able to access to the supplementary material. This is related to the definition of applied biases.
    The authors should specify a little more in the text how tha biases are applied in the simulated images. How "spurious observations due to steep topography near the source" have been simulated?

 With these remarks accounted for the paper is fully acceptable for Remote Sensing

Reviewer 3 Report

General comment

Hill and Nassar provide a comprehensive account of the impacts of satellite footprint size
and shape on the accuracy and precision of power plant CO2 emission estimates, taking
into account various biases. Additionally, they developed a method to average multiple
images of CO2 enhancements, and derived the required number of overpasses to quantify
annual emissions with a desired precision, which could have implications for the design
of future satellite missions.
The manuscript is well written with sufficient detail and explanation. The subject matter
is relevant to the scope of Remote Sensing. I have three major concerns that require
addressing. All other comments are minor and are mainly to do with clarifications or
corrections.

-----
Major comments:
1. In this manuscript, accuracy and precision in power plant CO2 emission estimate based
on satellite observations were evaluated for footprint sizes of 2–10 km, and it was
demonstrated that the smallest footprint sizes in this range (i.e., 2 km) have an advantage.
The impact of even smaller footprint size (less than 2 km) on the emission estimate was
only expected in Section 4 (P19, 392–396). However, many readers would like to know
which footprint size is practically appropriate for the emission estimate, and the
information is invaluable in determining an optimal footprint size. Therefore, I
recommend to adding similar investigations for footprint sizes less than 2 km (at least 1
km).
2. In this study, Gaussian plume model was used for simulating CO2 enhancements, and
CO2 emission was estimated using the Gaussian plume model. Although errors in CO2
emission estimate resulting from various biases were evaluated, deviation from Gaussian
plume model of actual plume was not taken into account in the error analysis, which could
be one of the largest error sources (e.g., Varon et al., 2018). This evaluation would be
made possible by simulating a plume using Weather Research and Forecasting model or
Large Eddy Simulation model and then by estimating CO2 emission using the Gaussian
plume model. If this is technically difficult, superposition of Gaussian plume model with
different atmospheric stability parameters may be another solution.
3. Authors mentioned that averaging multiple images is especially effective in estimating
CO2 emission from a weak source (P13, L285–292). Although, to carry out this method,
information on accurate wind direction is essential, it is difficult to estimate wind
direction from individual images for a weak emission source, as apparent in Figure 11.
Do you believe that wind direction from reanalysis data is accurate enough to perform
this method? Alternatively,

-----
Minor comments:
P2, L51–73:
It is determined that … (L55–58)
It is shown that … (L60–64)
We determined at least … (L68–69)
Although the results … (L69–73)
These four sentences are described in Conclusions and are not necessary in Introduction.
P3, L99–101: Please cite references.
P4, Figure 1: What is wind speed and atmospheric stability parameter used in the
simulations?
P8, L176–179: In association with just above comment, these sentences should be moved
into Section 2, because simulated figures (Figures 1–4) are already shown in Section 2.
Section 3.1:
Do you use “precision” and “uncertainty” in the same sense? These words should be
defined given that they are used in different senses, otherwise should be unified.
P16, L323–324: Does this sentence mean that any seasonal component was corrected
using the scale factors shown in Figure 13?
P18, L361: It is unclear what “daily emission variation” and “daily variability”
individually mean.
P19, L414 & P20, L452: In Section 3.2, an uncertainty in emission estimate from a weak
source (6 Mt yr-1) using 16 images is estimated to be 6.78%. Which value is correct?
In Supplementary materials
P4, Section 1.1.6: Because wind direction error and the related figure (Figure 8) are
described in the main text, this section is not necessary in supplementary material.
Figure S16: This figure is not referred in the main text.

-----
Technical corrections:
P2, L48: EPA → Environmental Protection Agency (EPA)
P9, L215: (Figure 5. → Figure 5.)
P11, L239: lost observations due to clouds shows → lost observations due to clouds
(figure S9) shows
P13, L286: For a source this small → For a smaller source
Figure 12: TThe label → The label
In Supplementary materials
P2, L19: 5% → 0.5%
P2, L21: 3% → 0.3%

Reference:
Varon, D.J.; Jacob, D.J.; McKeever, J.; Jervis, D.; Durak, B.O.; Xia, Y.; Huang, Y.
Quantifying methane point sources from fine-scale satellite observations of atmospheric
methane plumes. Atmospheric Measurement Techniques 2018, 11, 5673–5686.
